# Cis-Element Engineering Promotes the Expression of *Bacillus subtilis* Type I L-Asparaginase and Its Application in Food

**DOI:** 10.3390/ijms23126588

**Published:** 2022-06-13

**Authors:** Jiafeng Niu, Ruxue Yan, Juan Shen, Xiaoyu Zhu, Fanqiang Meng, Zhaoxin Lu, Fengxia Lu

**Affiliations:** College of Food Science and Technology, Nanjing Agricultural University, Nanjing 210095, China; 2021208017@stu.njau.edu.cn (J.N.); 2020108034@stu.njau.edu.cn (R.Y.); 2016027@njau.edu.cn (J.S.); zhuxiaoyu@njau.edu.cn (X.Z.); mfq@njau.edu.cn (F.M.); fmb@njau.edu.cn (Z.L.)

**Keywords:** L-asparaginase, *Bacillus subtilis*, promoter, dual-promoter systems, ribosome binding site, scale-up fermentation, acrylamide

## Abstract

Type I L-asparaginase from *Bacillus licheniformis* Z-1 (BlAase) was efficiently produced and secreted in *Bacillus subtilis* RIK 1285, but its low yield made it unsuitable for industrial use. Thus, a combined method was used in this study to boost BlAase synthesis in *B. subtilis*. First, fifteen single strong promoters were chosen to replace the original promoter P43, with PyvyD achieving the greatest BlAase activity (436.28 U/mL). Second, dual-promoter systems were built using four promoters (PyvyD, P43, PaprE, and PspoVG) with relatively high BlAase expression levels to boost BlAase output, with the engine of promoter PaprE-PyvyD reaching 502.11 U/mL. The activity of BlAase was also increased (568.59 U/mL) by modifying key portions of the PaprE-PyvyD promoter. Third, when the ribosome binding site (RBS) sequence of promoter PyvyD was replaced, BlAase activity reached 790.1 U/mL, which was 2.27 times greater than the original promoter P43 strain. After 36 h of cultivation, the BlAase expression level in a 10 L fermenter reached 2163.09 U/mL, which was 6.2 times greater than the initial strain using promoter P43. Moreover, the application potential of BlAase on acrylamide migration in potato chips was evaluated. Results showed that 89.50% of acrylamide in fried potato chips could be removed when combined with blanching and BlAase treatment. These findings revealed that combining transcription and translation techniques are effective strategies to boost recombinant protein output, and BlAase can be a great candidate for controlling acrylamide in food processing.

## 1. Introduction

L-asparaginase (EC 3.5.1.1) is a member of the amidohydrolase family that catalyzes the conversion of L-asparagine to L-aspartic acid and ammonia, and the molecular reaction scheme is shown in Figure 1 [1]. Because tumor cells lack the enzyme asparagine synthetase ability, the enzyme has received more attention for its efficiency in inhibiting tumor cell proliferation by depriving them of nutrition (L-asparagine), resulting in tumor cells starving to death, with little effect on healthy cells due to their enzyme asparagine synthetase ability [2]. The enzyme was also employed as a promising acrylamide-mitigating agent to manufacture acrylamide-free food products by eliminating asparagine (a significant precursor of acrylamide), with no modification in appearance, sensory qualities, flavor, and nutrition [3,4,5,6], in addition to its clinical application. As a result, novel L-asparaginases with good characteristics are urgently needed to be investigated.

Numerous L-asparaginases have been discovered in a variety of natural sources (microorganisms, animals, and plants), with microbial L-asparaginases emerging as the most promising source for large-scale L-asparaginase synthesis in healthcare and food industries [1,6,7,8]. However, the poor yield of L-asparaginase produced by these wild-type bacteria has hampered its widespread use. As heterologous recombinant strains can efficiently increase the expression level of interest proteins, numerous L-asparaginase genes have been successfully expressed in *Escherichia coli, Bacillus subtilis*, and *Pichia pastoris*, demonstrating that heterologous recombinant strains may rapidly boost the expression level of interest proteins. Among these expression systems, *B. subtilis* with GRAS (generally regarded as safe) status and strong secretion ability of recombinant proteins was the most suitable host for the production of proteins used in the food and pharmaceutical industry [9,10].

To increase the production of recombinant proteins in *B. subtilis*, which is the most intensively investigated Gram-positive bacterium, transcription and translation combination techniques have been applied. The strength of promoters that control the expression level of recombinant proteins is intimately related to transcriptional behavior [11]. In the past few decades, single strong promoter [12,13], dual-promoter [14], and triple-promoter [15,16] expression systems were created to improve transcriptional levels. The ribosome binding site (RBS) and spacer between the RBS and the start codon were also components that allowed gene expression to be regulated at the translational level [17,18,19,20]. Pang et al., for example, used a mix of promoter and RBS engineering methods to greatly improve pullulanase yield [21]. Previously, a type I L-asparaginase from *Bacillus licheniformis* Z-1 (BlAase) with good enzymatic characteristics was successfully produced and secreted in *B. subtilis* [22]. However, the yield of this enzyme was quite low, and thus hard to meet industrial demand. Therefore, the enzyme’s production needs to be improved to meet industrial demand.

In this study, a combination of transcription and translation methods was applied to boost BlAase output. The BlAase activity was 568.59 U/mL after single promoter screening, dual-promoter systems development, and promoter core regions (−35 and −10 boxes) optimization. Furthermore, RBS sequence change increased BlAase production at the translation level, with BlAase activity reaching 790.1 U/mL, which was 2.27 times greater than the original promoter P43-containing strain. Finally, the scale-up fermentation was carried out in a 10 L fermenter, and the BlAase activity was increased to 2163.09 U/mL, which was 6.2 times greater than the original promoter P43-containing strain. Furthermore, the acrylamide mitigation ability of fried potato chips with BlAase was also studied, and 89.50% acrylamide could be removed after blanching and BlAase treatment. These findings revealed a systematic technique for increasing recombinant protein production in *B. subtilis*.

## 2. Results and Discussion

### 2.1. Enhance BlAase Production with a Single-Initiator Subsystem

To date, numerous L-asparaginase genes have been successfully expressed in *E. coli, B. subtilis,* and *P. pastoris*. However, the L-asparaginase production yields in these recombinant strains were very limited, especially in food-grade hosts, hampered its industrial application [9,10]. Previously, we achieved expression and secretion of BlAase in recombinant *B. subtilis* for food safety, but the expression level was quite low, which was difficult to meet industrial needs. Thus, to increase BlAase production at the transcriptional level, different strong single promoters were selected to replace the original P43 promoter (Figure 2A). The relative strength of these promoters was determined using qRT-PCR. As shown in Figure 3B, the transcription strength of the PyxiE, Pylbp, PspoVG, PyvyD, and PamyE promoters was higher than that of the P43 promoter, which was 1.87-, 1.33-, 1.85-, 2.35-, and 2.17-fold, respectively. However, of these investigated promoters, only promoter PyvyD performed better, compared with promoter P43, with enzyme activity reaching 436.28 ± 17.37 U/mL (intracellular and extracellular enzyme activity reached 223.96 ± 10.64 and 212.32 ± 6.73 U/mL, respectively) after 72 h of cultivation at 37 °C, which was 1.25 times higher than that of the control strain with promoter P43 (348.75 ± 12.83 U/mL, intracellular and extracellular enzyme activity reached 190.96 ± 6.05 and 157.79 ± 6.78 U/mL, respectively) (Figure 3A). It is not surprising, since the promoter P43 was the most widely used promoter in *B. subtilis*, it has been used for the mediation of high-level expression of numerous recombinant proteins, and the strength of a single promoter was hardly stronger than that of P43 [12,23,24,25,26]. SDS–PAGE analyses (Figure 3C) also confirmed this result and were in good agreement with BlAase activity. It has been reported that even engines with the same promoter have different gene expression levels [27]. Compared with the constitutive promoter P43, the promoter PsrfA shows better performance in mediating the expression of green fluorescent protein (GFP) [28]. However, in this study, we observed the opposite result, that is, the production of BlAase under the control of the promoter P43 was much higher than that of the promoter PsrfA. This is reasonable because the expression level of recombinant proteins is affected by the transcription process, but also greatly depends on the translation rate [29,30,31,32,33].

### 2.2. Enhance BlAase Production with a Dual-Promoter System

According to reports, the dual-promoter system is one of the factors that affect the transcription intensity of recombinant proteins, and its optimization is considered to be an effective strategy to increase the yield of heterologous proteins [14,34]. Although BlAase yield was increased by optimizing individual promoters, the performance of single promoter systems was not ideal enough for industrial applications. Thus, in accordance with the characteristics of single promoters (Figure 4A), four promoters (PyvyD, P43, PaprE, and PspoVG) with relatively higher levels of BlAase expression were chosen to create systems with two promoters to further increase BlAase accumulation, and sixteen recombinant vectors were constructed mediated by a double-promoter upstream of BlAase (shown in Appendix A). To determine the expression level of these dual-promoter systems, both intracellular and extracellular BlAase activity were measured. According to the results shown in Figure 4A, dual-promoter systems performed ideal when nine of the sixteen dual-promoter systems (PaprE-P43, PyvyD-P43, PspoVG-P43, PaprE-PaprE, PyvyD-PaprE, PyvyD-PyvyD, P43-PyvyD, PaprE-PyvyD, and PspoVG-PyvyD) achieved higher yields compared to the parent P43 promoter. Among these nine systems with two promoters, the PaprE-PyvyD promoter reached the highest BlAase activity, up to 502.11 ± 23.51 U/mL (intracellular and extracellular enzymatic activity reached 134.57 ± 5.39 and 367.54 ± 18.12 U/mL, respectively), which was 1.44 times higher than BlAase-mediated activity from the original promoter P43. Previously, dual-promoter expression systems were also constructed to enhance the β-cyclodextrin glycosyltransferase (β-CGTase) expression level, and the dual promoter PHpaII-PamyQ′ showed the highest β-CGTase activity, which was only 1.3-fold higher than that of single promoter PamyQ′ [14]. To further confirm the above results, SDS–PAGE analysis of both extracellular and intracellular protein samples was performed, which largely corresponded to enzyme activity (Figure 4C).

In previous studies, the tandem promoter system was constructed by inserting different single promoters downstream of the superior promoter, with varying expression intensities [14,34]. Therefore, 16 dual-promoter systems were constructed with all possible sequences of four single strong promoters selected according to BlAase expression level. Compared with the Px-PspoVG dual-promoter system, Px-PyvyD achieved higher expression intensity (Figure 4A). This result may be due to the expression intensity of the dual-promoter system being highly dependent on the transcription strength of the promoter adjacent to the reporter gene [14]. In addition, qRT-PCR was performed to directly understand the transcriptional levels of BlAase in recombinant strains, and different dual-promoter system engines were applied to see how BlAase expression intensity was affected at transcriptional levels. As shown in Figure 4B, all of the transcriptional strengths of the tandem promoter systems were higher than that of the parental single strong promoter P43. The PaprE-PyvyD promoter with the highest level of BlAase expression, the level of transcription of which was more than 2.8 times higher than that of P43.

The above results indicated that the dual-promoter systems are an excellent strategy for improving the production yields of recombinant proteins at the transcription level.

### 2.3. Optimization of the Core Region of Promoter PyvyD to Increase Transcription Intensity

According to reports, the transcription strength of a promoter is directly related to its core region (−35 and −10 boxes), and optimizing these conserved regions was considered to be one of the important strategies to increase the yield of recombinant proteins [35,36]. Promoter PyvyD is a double-sigma factor-dependent promoter (σ^B−^- and σ^H−^-dependent promoter), and the consensus sequence is shown in Figure 5A [37]. However, these sequences do not exactly match the consensus sequence of the −35 and −10 boxes of the σ^B-^-and σ^H−^-dependent promoters in *B. subtilis*, respectively (Figure 5A). In addition, the latter promoter adjacent to the dual-promoter system reporter gene serves as a key promoter and plays a pivotal role in controlling the expression of the recombinant protein. Therefore, in order to further improve the transcriptional level of the dual-promoter system, the (−35 and −10 boxes of the downstream promoter PyvyD of the tandem promoter PaprE-PyvyD were changed to the corresponding consensus sequences separately or in combination, and three mutations were made (shown in Figure 5A). The results showed that mutant-1 and mutant-2 contained modifications of the (−35 and −10 regions of σ^B-^- and σ^H−^-dependent promoters, respectively. Expression levels of mutant-1 and mutant-2 reached 535.05 ± 33.35 and 539.53 ± 18.54 U/mL, compared with the original promoter (PaprE-PvvyD) (Figure 5B), 6.6% and 7.3% higher than wild type PaprE-PyvyD promoter, respectively. In addition, mutant-3, including a combination of consensus sequence optimizations, and BlAase activity achieved up to 568.59 ± 30.52 U/mL (Figure 5B), enhanced by 13.3% in contrast to promoter PaprE-PyvyD. However, the expression level of BlAase of mutant-1, mutant-2, and mutant-3 were 1.53, 1.55, and 1.63 times higher than that of the parental strain with promoter P43, respectively. Subsequently, the BlAase expression levels of three mutants were also confirmed by using SDS–PAGE (Figure 5D). In addition, the qRT-PCR analysis showed that the modification of the core region of the promoter PaprE-PyvyD directly enhanced the transcription intensity of BlAase (Figure 5C) and that the transcription intensity of mutant-1, mutant-2, and mutant-3 was 4.37, 4.15, and 4.86 times higher than that of promoter P43, respectively. The transcription intensity was much higher than the expression level of BlAase. This is not surprising, as the promoter is an effective control element for transcription, thereby influencing the mRNA abundance, while protein expression levels are directly controlled by the translation initiation rate. Similarly, several examples of disparity between mRNA abundance and protein expression levels have also been reported [12,15]. Moreover, it has also been reported that by optimizing the core region of the promoter to the consensus sequence of the recombinant strain, the yield of recombinant protein was significantly increased [35,36,38,39]. For example, Rao et al. reported that α-amylase activity was increased by 32.20% via optimizing sigma=factor binding sites (−10 and −35 boxes) of promoter PykzA-P43 [39]. This phenomenon might be due to the mRNA abundance reaching a threshold, and the increase in mRNA will not have a significant effect on protein expression level.

These results showed that optimization of the promoters of conserved regions can significantly improve the level of BlAase production.

### 2.4. Enhanced BlAase Production by Optimizing RBS

In the above section, the transcriptional strength of mutant-3 was increased by 70%, compared with the PaprE-PyvyD promoter, which was 4.86 times higher than that of the parental strain with the P43 promoter. However, the expression level of mutant-3 was only 1.63 times higher than that of the parental strain with the P43 promoter. This inconsistency in the relationship between the levels of transcription and translation can be caused by the low rate of translation initiation of BlAase. Thus, to further enhance the translational strength of mutant-3, 22 RBS sequences were selected to replace the original RBS sequence (RBS 0: GGGAGGCGTTCTTTG (shown in Figure 2A)). RBS 1-RBS 14 were strong RBS sequences for expressing endogenous or heterologous proteins in *B. subtilis* or *E. coli*, and RBS 15-RBS 22 were developed using the online tool “RBS Calculator v2.0” (https://salislab.net/software/, accessed on 2 April 2021) with a different translation initiation rate (shown in Appendix A). As shown in Figure 6A, the levels of BlAase production mediated by different RBS sequences were different. Compared with mutant-3, almost all strains containing the RBS sequence designed by “RBS Calculator v2.0” showed a lower expression intensity, except for RBS 18, which showed a comparable BlAase active (total BlAase activity reached 574.03 U/mL). These results may be due to the fact that RBS Calculator is a hypothesis and analysis software that cannot accurately predict protein expression levels [19,40]. In addition, compared with the original RBS sequence, 6 of 22 RBS sequences (RBS 1, RBS 3, RBS 7, RBS 10, RBS 13, and RBS 18) showed higher expression levels, and the total BlAase activity reached 790.1 ± 32.27 U/mL (intracellular and extracellular enzyme activities reached 221 ± 2.44 and 569.1 ± 14.74 U/mL, respectively). After being cultured at 37 °C for 72 h, RBS 10 was 1.39 times higher than mutant-3 but 2.27 times higher than the original strain containing the promoter P43. Similarly, based also on a report by Li et al., compared with the original strain, the L-asparaginase activity was enhanced by 209% after promoter screening and RBS optimization [40]. SDS–PAGE analysis also confirmed these results, which were in good agreement with the results of BlAase activity (Figure 6B).

The RBS sequence is a key factor in determining the translation initiation rate and protein expression level, and optimizing this region is considered an effective strategy to increase the yield of recombinant protein [17,41,42,43]. In order to better control the translation initiation rate and protein expression levels, the RBS calculator was developed and applied in the regulation of gene expression of recombinant strains [33]. Previous studies have used the RBS calculator to generate the RBS sequence library, the purpose of which was to optimize protein expression levels; consequently, the expression levels of proteins have been significantly improved [17,40].

The above results indicated that optimizing the RBS sequence is an effective method to increase the production of BlAase at the translation level.

### 2.5. BlAase Production in a 10 L Fermentation Machine

The BlAase activity was increased from 348.75 ± 12.83 to 790.1 ± 32.27 U/mL after improvement in transcription and translation levels, which was 2.27-fold higher than that of the parental strain containing the P43 promoter. However, the yield of this enzyme was still low, and industrial demand was difficult to meet. Thus, to further improve the levels of BlAase production, large-scale fermentation was performed to increase the cell density of *B. subtilis* contained with optimized expression elements in a 10 L fermenter. As shown in Figure 7A, intracellular BlAase was produced starting at 8 h and reached a maximum at 28 h (860.02 ± 93.87 U/mL). Extracellular BlAase activity was detected after 16 h of culture and reached a plateau at 1482.45 ± 62.79 U/mL after 36 h of culture. In addition, the total expression level after 36 h of cultivation reached 2163.09 ± 76.08 U/mL (intracellular and extracellular enzyme activity reached 680.64 ± 13.29 and 1482.45 ± 62.79 U/mL, respectively), which was 6.2 times higher than that of the original strain containing P43 promoter. The BlAase activity produced by high-density cell culture was higher than that of most microbial L-asparaginases expressed in *B. subtilis* [8,12,44,45]. As shown in Figure 7A, the L-asparaginase activity was strictly increased with cell growth in a 10 L fermenter, indicating that the cell density positively affected the yield of BlAase. Thus, increasing attention must be paid to the high cell density culture strategies to improve the levels of BlAase production in the future [12]. The above results will benefit the industrial production of the BlAase, and scale-up optimization of the fermentation conditions is required to further improve the production of BlAase in future studies.

### 2.6. Food Application of BlAase

Blanching is a pretreatment method that is applied in most vegetable and fruit processing industries. It is known that blanching will change the microstructure of the potato strips and increase the contact probability of asparaginase and asparagine [46]. Therefore, blanching combined with BlAase treatment was performed to reduce acrylamide content in fried potato chips. The effects of BlAase and blanching treatment on acrylamide elimination from fried potato chips were evaluated via LC–MS/MS (Appendix A). As shown in Figure 8, the acrylamide content of the control group of potato chips was 1.436 ± 0.036 mg/kg. After the blanching and BlAase treatment, the residual acrylamide contents were 0.817 ± 0.022 and 0.552 ± 0.011 mg/kg, respectively. Surprisingly, compared with the control group, the acrylamide content with the blanching and BlAase treatment decreased by 43.09% and 61.57%, respectively. After the combined blanching and BlAase treatment, the acrylamide content was merely 0.152 ± 0.027 mg/kg, indicating that 89.50% of acrylamide could be removed. As acrylamide content in fried food could be effectively removed by combined blanching and enzymatic treatment, simultaneous blanching and enzymatic treatment might greatly improve the efficiency of the treatment. Therefore, to expand the potential application of BlAase, its thermal stability should be optimized in future research in order to promote its development for use in food industries [45].

In previous studies, heat-treated potato slices caused a change in the microstructure of potato strips, such as the starch swells and cell-wall degradation; these changes might accelerate the diffusion of asparagine toward the L-asparaginase in the surrounding solution. Thus, blanching produces microstructure changes in potato tissue, which facilitates interaction between asparagine and L-asparaginase. However, when raw potato slices directly contact L-asparaginase, the potato microstructure limits interaction between asparagine and asparaginase [46].

The above results indicated that the combination of blanching and enzyme treatment is an efficient method to control acrylamide content in high-temperature processed foods.

## 3. Materials and Methods

### 3.1. Strains and Plasmids

The promoter sequences came from *B. subtilis* 168 (College of Food Science and Technology, Nanjing Agricultural University, Nanjing, China). The cloning and expression hosts were *E. coli* JM109 (College of Food Science and Technology, Nanjing Agricultural University, Nanjing, China) and *B. subtilis* RIK 1285 (TaKaRa, Dalian, China), respectively. The empty plasmid pP43NMK (a gift from Dr. Yunbin Lv, Jiangnan University, Wuxi, China) was utilized as an expression vector for L-asparaginase. BlAase expression was previously achieved using the vector pP43NMK-BlA-His (College of Food Science and Technology, Nanjing Agricultural University, Nanjing, China). L-asparagine was purchased from Aladdin (Shanghai, China). Kanamycin was obtained from Solarbio (Beijing, China). All other chemicals were of analytical grade.

### 3.2. Plasmid Construction

The expression vector pP43NMK-BlA-His constructed previously was used as a backbone for further study. To boost BlAase transcription in *B. subtilis*, 15 strong promoters (Appendix A) were cloned and utilized in place of the original promoter P43. Using the ClonExpression II One Step Cloning Kit (Vazyme, Nanjing, China), the various promoter segments were cloned from *B. subtilis* 168 with matched primer pairs (Appendix A) and fused to the vector pP43NMK-BlA-His, linearized via polymerase chain reaction (PCR) to replace promoter P43. Dual-promoter expression systems were created using a similar manner.

The mutation was introduced with appropriate primer pairs to modify the core region of the PyvyD promoter (Appendix A). In addition, 22 RBS and spacer sequences (Appendix A) were constructed and used to replace the original sequence to improve BlAase production at the translation stage. Figure 2 depicts a promoter and RBS engineering schematic. All sequenced vectors were transformed into *B. subtilis* RIK 1285 competent cells for BlAase expression using the approach outlined by Dubnau [47].

### 3.3. Expression of the BlAase in B. subtilis

All modified strains were inoculated into 250 mL shake flasks contained with 50 mL fermentation medium (45× *g* sucrose, 12× *g* maize starch, 15× *g* peptone, 0.8× *g* urea, 2.612× *g* K_2_HPO_4_·3 H_2_O, 2.041× *g* KH_2_PO_4_, 1.845× *g* MgSO_4_·7 H_2_O, 3× *g* NaCl, 1× *g* L-asparagine, pH = 7.5) and 50× *g* of kanamycin and then cultured at 37 °C and 200 rpm for 72 h. The culture was centrifuged at 8000 rpm and 4 °C for 10 min after fermentation, and the supernatant was used as an extracellular sample. The precipitate was resuspended in 50 mM Na_2_HPO_4_-NaH_2_PO_4_ (pH = 8.0) and 300 mM NaCl in the binding buffer. The resuspended cells were then ultrasonically lysed for 20 min at 4 °C. Centrifugation at 4 °C for 10 min to remove the lysed cells and the supernatant was used as an intracellular sample. Sodium dodecyl sulfate-polyacrylamide gel electrophoresis (SDS–PAGE) was used to distinguish intracellular and extracellular proteins, as described by Leammli [48]. Coomassie Brilliant Blue G250 was used to stain the isolated proteins.

### 3.4. RNA Extraction and qRT-PCR

Sun et al. proposed a method for determining the transcription level of BlAase in *B. subtilis* [17]. Total RNA was extracted using the TransZol Up kit (TransGen, Beijing, China) according to the manufacturer’s instructions, and Nanodrop 2000 (Thermo Scientific, Waltham, MA, USA) was used to determine the extracted RNA concentration and purity. The HiScript^®^ II Q RT SuperMix for qPCR kit (Vazyme, Nanjing, China) was reverse transcribed RNA into cDNA according to the manufacturer’s instructions. The level of BlAase transcription was determined using quantitative real-time PCR (qRT-PCR), with cDNA as the template and 16S rDNA as the internal control. Livak et al. described a 2^−^^△△Ct^ method for calculating the relative expression levels [49].

### 3.5. Measurement of BlAase Activity

The amount of ammonia liberated from the hydrolysis of L-asparagine was calculated to determine L-asparaginase activity, and the BlAase activity was determined using the method described in our earlier study [22]. The amount of enzyme necessary to liberate 1 μmol of ammonia per mL and per min at pH 8.0 at 37 °C was established as one unit (U) of L-asparaginase activity.

### 3.6. Production of BlAase in a 10 L Fermenter

To enhance the production of BlAase and better industry application, scale-up fermentation was carried out in a 10 L fermenter using the 4 L fermentation medium stated above to improve BlAase production and its industrial use. As a seed culture, the modified strain with the BlAase gene under the control of the dual-promoter PaprE-PyvyD and RBS 10 was cultivated in fermentation media. When the cells reached the mid-log phase, the seed culture was injected into a 10 L fermenter (5 percent inoculum). The fermentation conditions are as follows: The stirring speed was controlled to 350–500 rpm to maintain the dissolved oxygen level above 20%; the temperature was maintained at 37 °C, and ammonia water was added to maintain the pH at 7.0–7.5. Samples were collected every 4 h to measure enzyme activity.

### 3.7. Reduction in Acrylamide in Potato Chips

#### 3.7.1. Sample Preparation

The sample was prepared according to the method described by Zhang [50], with slight modifications. Briefly, potatoes were washed, peeled, and cut into 2 mm slices. The potato slices were rinsed in distilled water to remove starch adhering to the surface. Then, they were pretreated before frying with various methods described below:
(I)They were rinsed in distilled water without any further processing at 37 °C for 2 h;(II)Potato slices were blanched at 85 °C for 10 min and then rinsed in distilled water at 37 °C for 2 h;(III)They were immersed in a 40 U/mL BlAase solution at 37 °C for 2 h;(IV)They were blanched at 85 °C for 10 min and then immersed in a 40 U/mL BlAase solution at 37 °C for 2 h.

The pretreated samples were fried at 180 °C for 5 min. The fried samples were cooled and dried at room temperature.

#### 3.7.2. Acrylamide Determination

The extraction and detection of acrylamide from fried potato slices were performed according to the method described by Chi [51]. Briefly, 10× *g* of fried potato slices were washed with 50 mL of n-hexane three times to eliminate oil. Then, 25 mL of deionized water, 25 mL of acetonitrile, 5 g of NaCl, and 20× *g* of magnesium sulfate were added to the fried potato chip samples to extract acrylamide. The samples were centrifuged at 10,000 rpm for 10 min to collect the acetonitrile layer, which contained the acrylamide. Subsequently, the acetonitrile layer was dried via rotary evaporation and redissolved with 1 mL of deionized water. Finally, the resuspended liquids were filtered through a 0.22 μm microporous membrane for further analysis.

The acrylamide content of the samples was determined via liquid chromatography–tandem mass spectrometry (LC–MS/MS) using a Kinetex F5100A (2.1 × 100 mm, 2.6 μm) chromatographic column. The samples were eluted at 30 °C with a mobile phase of 90% water and 10% methanol (*v*/*v*) at a flow rate of 0.25 mL/min. Acrylamide was detected at ions *m*/*z* 72.1 and m/z 54.9 in electrospray ionization source positive-ion mode.

### 3.8. Statistical Analysis

All experiments of this paper were repeated three times, and the values are means of three repetitions ± standard deviation. All statistical analyses were carried out using Origin 2021.

## 4. Conclusions

The BlAase has been effectively produced and secreted in *B. subtilis*, but its industrial applicability has been hampered by its exceedingly low yield. The BlAase production yield was improved in this study by combining transcription and translation approaches, such as single strong promoter screening, dual-promoter systems building, promoter core region alteration (−35 and −10 boxes), and RBS substitution. The overall BlAase activity arrived in a shake flask after engineering the expression components was 790.1 U/mL, which was 2.27 times greater than the parental strain with promoter P43. High-density cell culture was used in a 10 L fermenter to raise BlAase production yield and improve industrial applications. After 36 h of cultivation, the BlAase activity was 2163.09 U/mL, which was 6.2 times greater than the parental strain containing promoter P43. Moreover, a combined strategy of blanching and enzyme treatment was employed to control acrylamide levels in fried potato chips, and the acrylamide content decreased by 89.50%, compared with the control group. These findings revealed a systemic technique for increasing recombinant protein production in *B. subtilis*. In future research, we aim to improve the expression strength of BlAase by optimizing fermentation conditions; thus, BlAase will serve as a promising enzyme for applications in the food processing industry.

## Figures and Tables

**Figure 1 ijms-23-06588-f001:**
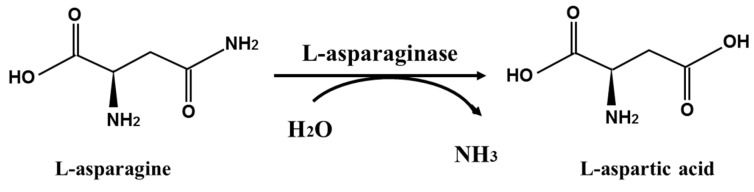
The molecular reaction scheme of L-asparaginase.

**Figure 2 ijms-23-06588-f002:**
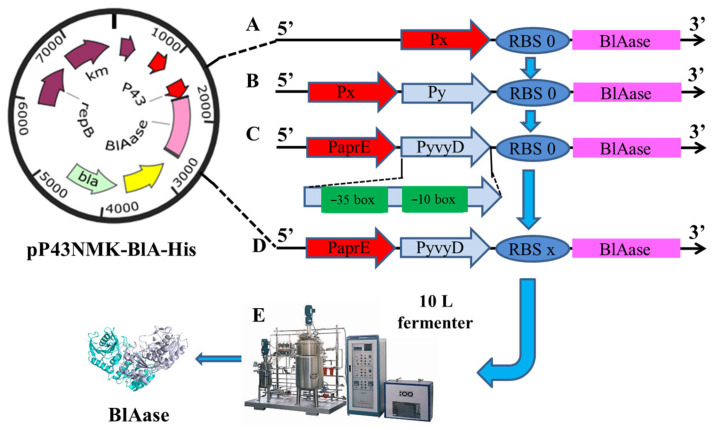
Vector construction with improved expression elements: (**A**) replacing the P43 promoter with a single strong promoter; (**B**) schematic diagram of the construction of dual-promoter systems; (**C**) optimization of core regions of the PyvyD promoter; (**D**) modification of RBS sequence from BlAase; (**E**) high-cell density culture in a 10 L fermenter. Px and Py represent different promoters; RBSx represents different RBS sequences; the −35 box and −10 box represent the core regions of the promoter.

**Figure 3 ijms-23-06588-f003:**
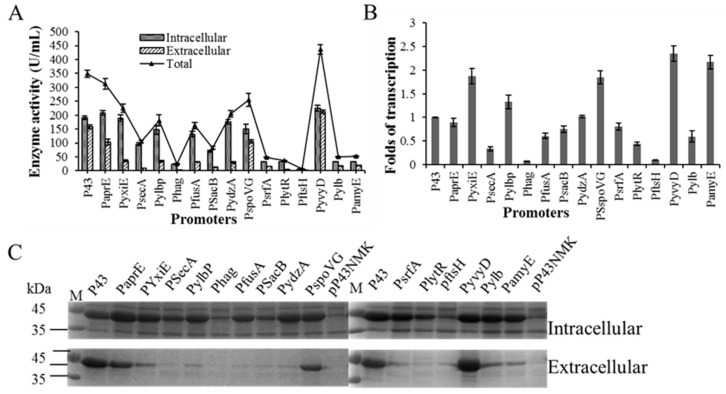
BlAase transcription and translation levels mediated by distinct single promoters: (**A**) BlAase expression levels with different single promoter engines; (**B**) BlAase transcription levels with different single promoter engines; (**C**) BlAase SDS–PAGE analysis with different single promoter engines.

**Figure 4 ijms-23-06588-f004:**
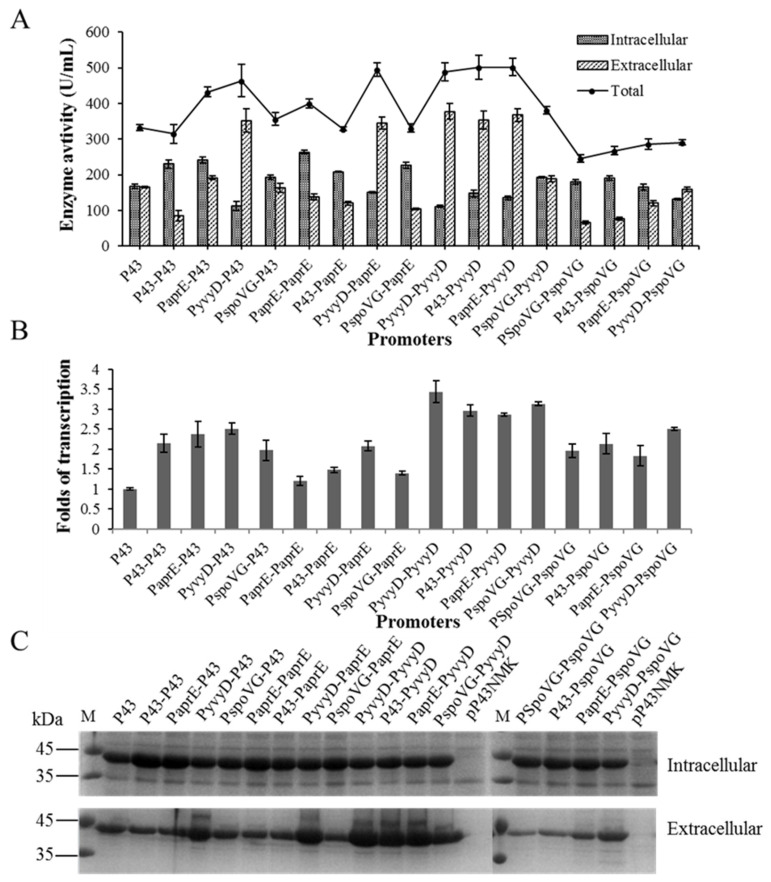
BlAase transcription and translation mediated by various dual-promoter systems: (**A**) BlAase expression levels with different dual-promoter systems’ engines; (**B**) BlAase transcription levels with different dual-promoter systems’ engines; (**C**) BlAase SDS–PAGE analysis with different dual-promoter systems’ engines.

**Figure 5 ijms-23-06588-f005:**
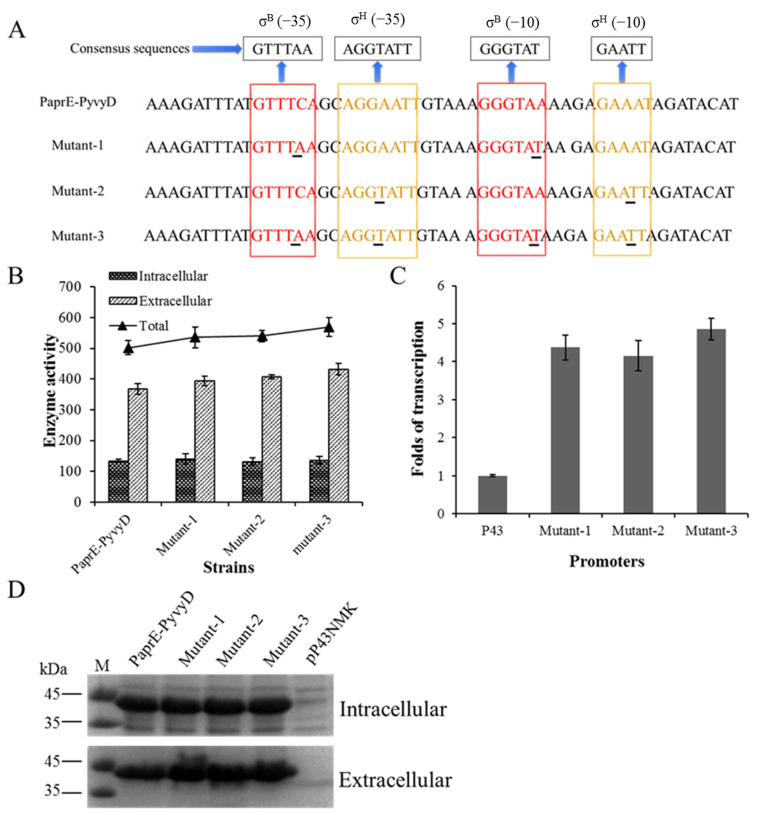
The core region modified promoter PyvyD regulates BlAase transcription and translation: (**A**) nucleotide sequences of the core region of the PyvyD promoter; (**B**) the transcription level of BlAase with the mediation of different mutants; (**C**) BlAase expression level with the mediation of different mutants; (**D**) BlAase SDS–PAGE analysis with the engine of different mutants.

**Figure 6 ijms-23-06588-f006:**
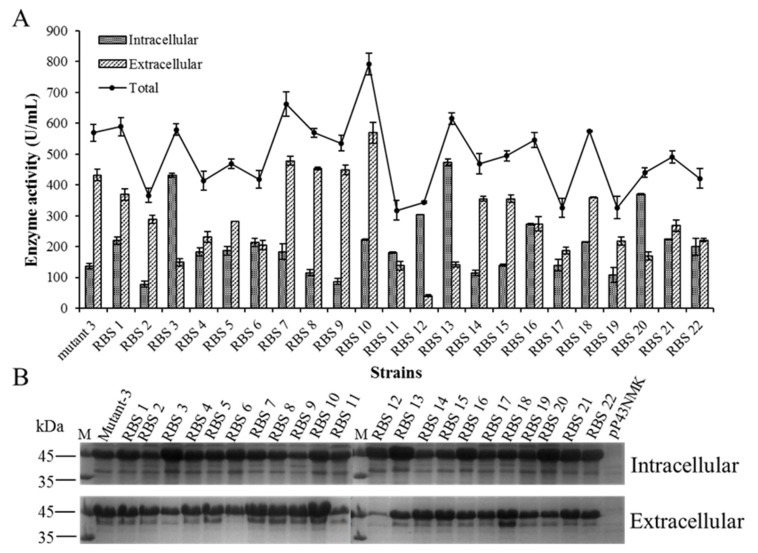
BlAase expression levels as a function of RBS sequences mediation: (**A**) BlAase expression levels with different RBS sequences as mediators; (**B**) BlAase SDS–PAGE analyses with different RBS sequences as mediators.

**Figure 7 ijms-23-06588-f007:**
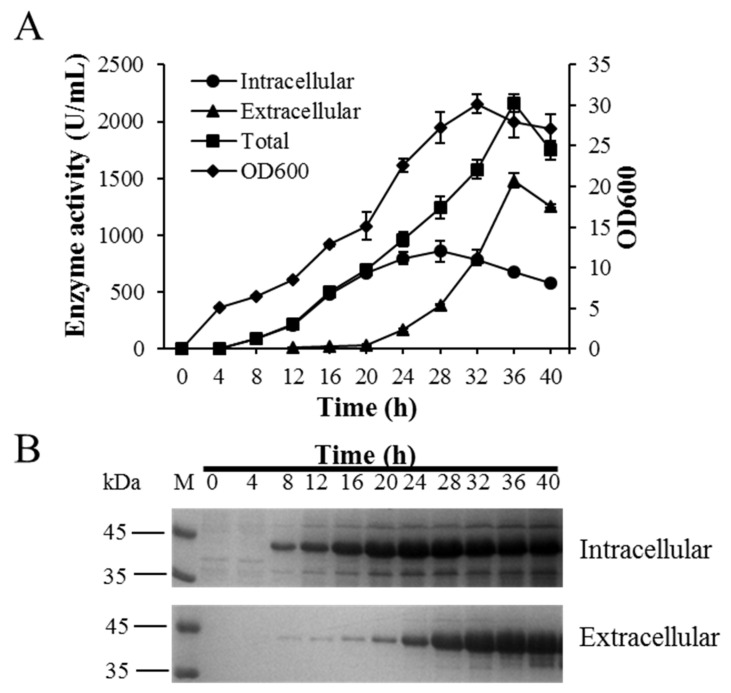
BlAase production in a 10 L fermenter: (**A**) BlAase activity and cell development over time in a 10 L fermenter; (**B**) BlAase SDS–PAGE analysis in a 10 L fermenter.

**Figure 8 ijms-23-06588-f008:**
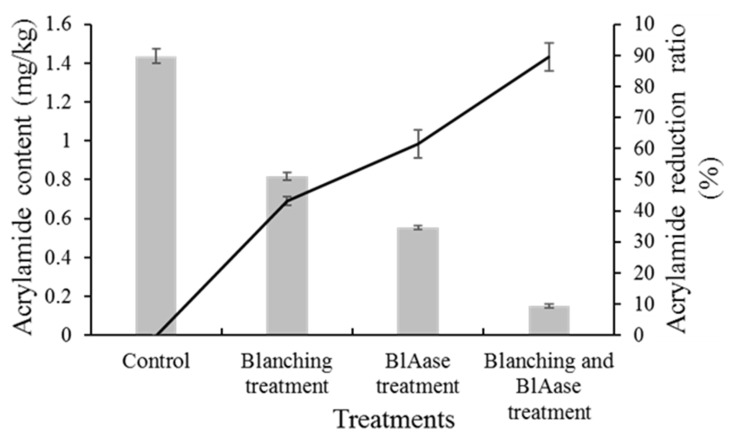
Acrylamide levels in potato chips subjected to different treatments.

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
