# Peer review of "Cis-Element Engineering Promotes the Expression of Bacillus subtilis Type I L-Asparaginase and Its Application in Food"

_ijms, 2022, doi:10.3390/ijms23126588_

Round 1

Reviewer 1 Report

In this article, authors describe their promoter and RBS engineering studies in order to enhance expression levels of an asparaginase and demonstrate a 10-L scale fermentation with their optimized gene constructs. Moreover, an application of the enzyme towards reduction of acrylamide content in food also has been shown. Since the enzyme can be useful for industrial applications, its efficient and large scale expression and production is desired. Thus this study is of importance and suitable for publication in IJMS. However, some major improvements should be made, as I pointed out below:

 1)      There are many writing errors, including grammar errors, typos, wrong sentence structure etc. throughout the text. I have pointed out some of them below. I suggest the authors to go through the manuscript carefully to correct those kinds of language errors and maybe get help from native speakers.

-Line 38 – better word choice can be made

-line 46 – recombinant(s) – “s” should be omitted

-line 46-47 – sentence should be revised

-line 53 “as the result” should be replaced

-lines 91, 92 and 93 – revise the sentences, not meaningful

-line 142 – is it “engine intensity” or “gene intensity”?

-line 143- “In addition, perform qRT-PCR to directly….” – correct sentence

-lines 147-149 – verb is missing in the sentence

-line 200 – conjunction is missing before “B1Ase”

-line 211 – “compared with” – correct

-line 260 – revise sentence for better readability

-line 301-302 – subscripts for chemical formulas

 2)      In the introduction, a molecular reaction scheme of the enzyme (showing substrate and product) should be included.

3)      The authors should state how many times the experiments were repeated. I can see error bars on the graphs, but it is not mentioned what was the number of repetitions. Moreover, when the numbers are reported throughout the text, the errors (standard error or deviation) should also be given. For example in line 176.

4)      Authors should compare and discuss their results with other studies a little bit more, in order to indicate that the obtained increase levels in this study are significant. For example, it will be good to convince reader how significant are 1.5-2 fold or 3-4-fold increases in expression/translation/transcription.

5)      There should be more discussion on why the increases in transcription levels are not reflected in translation and enzyme activity, especially in section 2.3.

6)      In section 2.6., “blanching” treatment should be briefly defined/described, since it may not be immediately apparent to researchers in different fields.

7)      In line 207, the meaning of the following sentence is not very clear: “Almost all strains contain the RBS sequence designed by “RBS Cal-207 culator v2.0”. Please clarify.

8)      What is the volume of regular fermentations, and what are the exact conditions – are they shake flasks? This should be clearly given in order to compare it with 10 L fermentation.

Author Response

RE: ijms-1755194

     Thank you very much for the valuable comments concerning our manuscript entitled “Cis-elements engineering promote the expression of Bacillus subtilis TYPE I L-asparaginase and its application in food” (Manuscript ID: ijms-1755194). The comments are very helpful for revising and improving our manuscript. We have carefully gone through the comments of reviewers and believe each of the concerns has been thoroughly addressed in the revised manuscript. Please refer to the attached Response to Reviewers for a point-by-point response. All revisions we made have been marked in red in the manuscript. We hope the manuscript will now meet with your approval.

We are grateful for the editor and reviewers’ input and hard work and believe the paper has been strengthened as a result of addressing their comments.

Thank you very much for your consideration.

Sincerely,

Fengxia Lu

College of Food Science & Technology, Nanjing Agricultural University

Nanjing 210095, China

Response to Reviewers

Reviewer 1

1) There are many writing errors, including grammar errors, typos, wrong sentence structure etc. throughout the text. I have pointed out some of them below. I suggest the authors to go through the manuscript carefully to correct those kinds of language errors and maybe get help from native speakers.

-Line 38 – better word choice can be made

Response: Thank you for your valuable comments and suggestions regarding the paper. We have modified it. (Line 39-40).

-line 46 – recombinant(s) – “s” should be omitted

Response: Thank you for your valuable comments and suggestions. We have omitted “s” of recombinants. (Line 50).

-line 46-47 – sentence should be revised

Response: Thank you for your valuable comments and suggestions regarding the paper. We have revised the sentence. (Line 47-54).

-line 53 “as the result” should be replaced

Response: Thank you for your valuable comments and suggestions. We have used “in the past decades” to replace “as the result”. The sentence was modified to “In the past decades, single strong promoter, dual-promoter, and triple-promoter expression systems were created to improve transcriptional levels.”. (Line 59).

-lines 91, 92 and 93 – revise the sentences, not meaningful

Response: Thank you for your valuable comments and suggestions. We apologize that we did not make it clear. We have revised the sentences. (Line 99-102).

 -line 142 – is it “engine intensity” or “gene intensity”?

Response: Thank you for your comments. In this sentence, “engine intensity” was referred as the transcription strength of promoters, and we have modified the sentence to “This result may be due to the expression intensity of the dual promoter system was highly dependent on the transcription strength of the promoter adjacent to the reporter gene”. (Line 156-157).

 -line 143- “In addition, perform qRT-PCR to directly….” – correct sentence

Response: Thank you for your valuable comments and suggestions. We apologize that we did not make it clear. We have corrected the sentences. (Line 157-160).

 -lines 147-149 – verb is missing in the sentence

Response: Thank you for your comments. We have modified the sentence. (Line 157-160).

-line 200 – conjunction is missing before “B1Ase”

Response: Thank you for your comments and suggestions. We have used “of” to conjunct BlAase. (Line 227).

 -line 211 – “compared with” – correct

Response: Thank you for your comments. We have corrected “compare with” to “compared with”. (Line 239).

 -line 260 – revise sentence for better readability

Response: Thank you for your comments and suggestions. We have modified the sentences. (Line 296-300).

 -line 301-302 – subscripts for chemical formulas

Response: Thank you for your comments and suggestions. We have corrected the errors. (Line 347-351).

Besides, we have also corrected the language errors in the revised manuscript from native speakers. (Line 8-9, 22, 38, 64-66, 79, 129, 135, 142, 165, 175, 208, 234, 247, 255, 337, 353, 355).

2) In the introduction, a molecular reaction scheme of the enzyme (showing substrate and product) should be included.

 Response 2: Thanks for your suggestion and comment. We have added a molecular reaction scheme of L-asparaginase in the introduction. Moreover, the figure number was also changed in revised manuscript. (Line 90-91, 99. 103, 113, 120, 130, 136, 149, 155, 161, 169, 178, 180, 186-187, 190, 193, 197, 199, 217, 229, 233, 248, 260, 270, 278, 286, 285, 318, 342).

Figure 1. The molecular reaction scheme of L-asparaginase. (Line 30-31, 41-42).

3) The authors should state how many times the experiments were repeated. I can see error bars on the graphs, but it is not mentioned what was the number of repetitions. Moreover, when the numbers are reported throughout the text, the errors (standard error or deviation) should also be given. For example in line 176.

Response 3: Thanks for your suggestion and comment. We apologize that we did not make it clear. In this paper, each experiment was repeated for three times, and the values are means of three repetitions ± standard deviation. For statistical analysis, we have added the statement as the new section 3.8 to the revised Materials and Methods. (Line 419-422). Moreover, the standard deviation of enzyme activities has also been given. (Line 95-96, 98-99, 140-142, 189, 241-242, 264, 271-272, 274-275, 297, 301).

4) Authors should compare and discuss their results with other studies a little bit more, in order to indicate that the obtained increase levels in this study are significant. For example, it will be good to convince reader how significant are 1.5-2 fold or 3-4-fold increases in expression/translation/transcription.

 Response 4: Thank you for your valuable comments and suggestions regarding the paper. In order to indicate the significance of the obtained increase levels in this study, we have compared and discussed our results with other studies in the revised manuscript. (Line 143-147, 209-213, 245-247).

5) There should be more discussion on why the increases in transcription levels are not reflected in translation and enzyme activity, especially in section 2.3.

Response 5: Thank you for your valuable comments and suggestions regarding the paper. As we known, the expression level of recombinant proteins is not only affected by the transcription process, but also greatly depends on the translation rate. Thus, the increases in transcription levels might not reflected in translation and enzyme activity. Moreover, we have also discussed this in the revised manuscript. (Line 201-206, 209-213).

6) In section 2.6., “blanching” treatment should be briefly defined/described, since it may not be immediately apparent to researchers in different fields.

 Response 6: Thank you for your valuable comments and suggestions. We apologize that we did not make it clear. We have briefly described the blanching treatment in section 2.6. Blanching is a pretreatment method which is applied in most vegetable and fruits processing industry. It is known that blanching will change the microstructure of the potato strips and increase the contact probability of asparaginase and asparagine. Therefore, blanching combined with BlAase treatment was performed to reduce acrylamide content in fried potato chips. (Line 289-293).

7) In line 207, the meaning of the following sentence is not very clear: “Almost all strains contain the RBS sequence designed by “RBS Cal-207 culator v2.0”. Please clarify.

 Response 7: Thanks for your comment. We modified the sentence by replacing “Almost all strains contain the RBS sequence designed by “RBS Calculator v2.0”. Compared with mutant-3, the expression intensity is lower, except for RBS 18 which has comparable BlAase active (total BlAase activity reached 574.03 U/mL).” with “Compared with mutant-3, almost all strains contain the RBS sequence designed by “RBS Calculator v2.0” showed a lower expression intensity, except for RBS 18, which showed a comparable BlAase active (total BlAase activity reached 574.03 U/mL).”.  (Line 234-237).

 8) What is the volume of regular fermentations, and what are the exact conditions – are they shake flasks? This should be clearly given in order to compare it with 10 L fermentation.

 Response 8: Thank you for your valuable comments and suggestions regarding this paper. We apologize that we did not make it clear. The regular fermentation was performed in 250 mL shake flasks contained with 50 mL fermentation medium, and the cultural conditions were at 37 °C and 200 rpm for 72 hours. Moreover, we modified the sentence in section 3.3 by replacing “All modified strains were cultivated for 72 hours in fermentation medium (45 g/L sucrose, 12 g/L maize starch, 15 g/L peptone, 0.8 g/L urea, 2.612 g/L K2HPO4·3 H2O, 2.041 g/L KH2PO4, 1.845 g/L MgSO4·7 H2O, 3 g/L NaCl, 1 g/L L-asparagine, pH=7.5) containing 50 µg/mL of kanamycin at 37 °C and 200 rpm.” with “All modified strains were inoculated into 250 mL shake flasks contained with 50 mL fermentation medium (45 g/L sucrose, 12 g/L maize starch, 15 g/L peptone, 0.8 g/L urea, 2.612 g/L K2HPO4·3 H2O, 2.041 g/L KH2PO4, 1.845 g/L MgSO4·7 H2O, 3 g/L NaCl, 1 g/L L-asparagine, pH=7.5) and 50 µg/mL of kanamycin and then cultured at 37 °C and 200 rpm for 72 hours.”. (Line 347-351).

Reviewer 2 Report

Dear Authors,

Thank you for your interesting manuscript. The manuscript is well prepared for both data and writing. Therefore, I strongly comment about accepting for publication after minor corrections.

Minor comments:

In conclusion, there is more discussion about limits and future work of this study which would be improved the quality of the manuscript.

For experiments, how many times repeat the experiments? This information should be confirmed and added to the manuscript.

There are a number of typos in the manuscript (e.g. there was a mix of "a space" between number and unit; temperature symbol (lines 326, 336...); font ( lines 350, 352...). Please thoroughly check and correct these typos. 

Author Response

RE: ijms-1755194

     Thank you very much for the valuable comments concerning our manuscript entitled “Cis-elements engineering promote the expression of Bacillus subtilis TYPE I L-asparaginase and its application in food” (Manuscript ID: ijms-1755194). The comments are very helpful for revising and improving our manuscript. We have carefully gone through the comments of reviewers and believe each of the concerns has been thoroughly addressed in the revised manuscript. Please refer to the attached Response to Reviewers for a point-by-point response. All revisions we made have been marked in red in the manuscript. We hope the manuscript will now meet with your approval.

We are grateful for the editor and reviewers’ input and hard work and believe the paper has been strengthened as a result of addressing their comments.

Thank you very much for your consideration.

Sincerely,

Fengxia Lu

College of Food Science & Technology, Nanjing Agricultural University

Nanjing 210095, China

Response to Reviewers

Reviewer 2

  1. In conclusion, there is more discussion about limits and future work of this study which would be improved the quality of the manuscript.

Response 1: Thank you for your valuable comments and suggestions. We have showed more discussion about limits and future work in revised manuscript. (Line 83-86, 143-147, 201-206, 209-213, 278-282, 302-306).

  1. For experiments, how many times repeat the experiments? This information should be confirmed and added to the manuscript.

 Response 2: Thanks for your suggestion and comment. We apologize that we did not make it clear. In this paper, each experiment was repeated for three times, and the values are means of three repetitions ± standard deviation. For statistical analysis, we have added the statement as the new section 3.8 to the revised Materials and Methods. (Line 419-422).

  1. There are a number of typos in the manuscript (e.g. there was a mix of "a space" between number and unit; temperature symbol (lines 326, 336...); font ( lines 350, 352...). Please thoroughly check and correct these typos.

 Response 3: Thanks for your suggestion and comment. We have thoroughly check and correct these typos in revised manuscript. (Line 8-9, 22, 38, 50, 64-66, 79, 129, 135, 142, 165, 175, 208, 234, 247, 255, 337, 351, 353, 355, 374, 385).
